# Explore the Value of Multi-Parameter MRI in Non-Invasive Assessment of Prognostic Risk and Oxford Classification in Children with IgAN or IgAVN

**DOI:** 10.3390/bioengineering11080750

**Published:** 2024-07-24

**Authors:** Zhouyan Liao, Guanjie Yuan, Kangwen He, Shichao Li, Mengmeng Gao, Ping Liang, Chuou Xu, Yu Zhang, Zhen Li

**Affiliations:** 1Department of Radiology, Tongji Hospital, Tongji Medical College, Huazhong University of Science and Technology, Wuhan 430030, China; 15271924784@163.com (Z.L.); ygjforever98@163.com (G.Y.); kangwen_he@163.com (K.H.); 15172541672@163.com (S.L.); gaomm22@hust.edu.cn (M.G.); pinglianglp@163.com (P.L.); xchuou@hust.edu.cn (C.X.); 2Department of Paediatrics, Tongji Hospital, Tongji Medical College, Huazhong University of Science and Technology, Wuhan 430030, China

**Keywords:** immunoglobulin A nephropathy, immunoglobulin A vasculitis with nephritis, blood oxygenation level-dependent-magnetic resonance imaging, ZOOMit diffusion-weighted imaging, Oxford classification

## Abstract

Purpose: To explore the Oxford classification and prognostic risk stratification of the non-invasive evaluation of immunoglobulin A nephropathy (IgAN) or immunoglobulin A vasculitis with nephritis (IgAVN) in children using multiparametric magnetic resonance imaging (MRI). Materials and Methods: Forty-four children diagnosed with IgAN or IgAVN were included. Patients with 80-month risk scores >10% were categorized as the high-risk group, while others constituted the low-risk group. The T2* and apparent diffusion coefficient (ADC) values of the renal cortex and medulla were measured. Clinical and pathological parameters were also assessed. Univariate and multivariate logistic regression analyses were performed to identify the indicators associated with the high-risk group. Receiver operating characteristic (ROC) curves were drawn and the areas under the curve (AUCs) were calculated to evaluate the diagnostic performance variables for differentiating the high-risk group from the low-risk group. Results: Only the T2*Cortex and mean arterial pressure (MAP) were independently reliable in both the univariate and multivariate analyses. The AUCs for differentiating the high-risk group from the low-risk group of T2*Cortex, MAP, and their combination model were 0.907, 0.881, and 0.947, respectively. Conclusions: Multiparametric MRI parameters, especially T2* values, could be used as new biomarkers to provide a new dimension in chronic kidney disease-related research and could play an important role in the non-invasive prognosis of children with IgAN or IgAVN.

## 1. Introduction

Immunoglobulin A nephropathy (IgAN) and immunoglobulin A (IgA) vasculitis with nephritis (IgAVN) are the most common primary and secondary forms of pediatric glomerulonephritis [1]. The spectrum of biopsy-confirmed renal diseases in Chinese children shows that IgAVN accounts for 29.17% of cases, while IgA nephropathy accounts for 22.70% of cases [2]. They are different manifestations of IgA-mediated inflammatory syndromes with a common pathogenesis and similar renal morphological and immunohistochemical features [3]. Many children with IgAN or IgAVN eventually progress to end-stage kidney disease (ESKD) in adulthood [4,5]. Therefore, it is necessary to accurately evaluate the long-term prognostic risk in children with IgAN or IgAVN.

The SHARE initiative classification provides evidence-based international advice on the diagnosis and treatment of pediatric IgAVN [6]. However, whether the classification of the severity of IgAVN using the SHARE initiative can effectively predict the prognosis of children remains to be further explored. The modified Oxford classification, recommended by the IgA Nephropathy Classification Working Group to predict the prognosis of IgAN, assesses mesangial hypercellularity (M), endothelial hypercellularity (E), segmental sclerosis or adhesion (S), interstitial fibrosis or tubular atrophy (T), and crescent (C) scores [7]. Compared with urinary protein, it can better predict the prognosis of IgAVN [8]. However, percutaneous kidney biopsy for the Oxford classification is not only an invasive procedure with risks of complications, such as pain, bleeding, and infection [9], but it is also difficult to repeat for longitudinal monitoring. Therefore, an effective non-invasive method is urgently needed to evaluate and monitor the progress of IgAN and IgAVN.

Blood oxygenation level-dependent magnetic resonance imaging (BOLD-MRI) is a very valuable non-invasive tool for evaluating renal blood oxygen levels without contrast agents [10]. A decrease in T2* (apparent spin–spin relaxation time) values suggests an increase in deoxyhemoglobin and kidney hypoxia. Diffusion-weighted imaging (DWI) is a tool that has been proven in many studies to be able to non-invasively assess renal function and renal fibrosis by reflecting the apparent diffusive movement of water molecules in kidney tissue [11,12]. However, most of the studies have been performed in adults with CKD [13], whereas functional MRI has rarely been used in pediatric patients with CKD [14]. Moreover, recent studies have focused on the correlation between T2* or ADC and renal function in children [15,16], and little attention has been paid to the ability of MRI to predict the prognosis of pediatric patients with CKD.

The aim of this study was to evaluate the ability of BOLD-MRI and DWI to non-invasively assess the Oxford classification and prognostic risk stratification and to identify the predictable clinical and MRI parameters in children with IgAN or IgAVN.

## 2. Methods

### 2.1. Patients

This retrospective study adhered to the Declaration of Helsinki’s rule of ethics and was approved by the ethics committee of Huazhong University of Science and Technology, Tongji Hospital, Tongji Medical College. The requirement for informed consent was waived. From January 2020 to December 2022, 91 children were pathologically diagnosed with IgAN or IgAVN in the Department of Pediatrics in our hospital and were enrolled in the study. Of them, 47 children were excluded because of a lack of MRI images or poor image quality. Finally, 44 children (34 males and 10 females) were included in this study.

### 2.2. Clinical and Pathological Information

For all the pediatric patients, we collected data including height, weight, blood pressure, urea, serum creatinine (Scr), uric acid, HCO_3_−, Cystatin C, the 24 h-urinary protein (Upro), the 24 h-urinary protein/creatinine ratio (PCR), urinary β2-microglobulin (β2-MG), urinary occult blood, and the neutrophil/lymphocyte ratio (NLR) within 7 days before the MRI examination from the medical record system of Tongji Hospital. In addition, the medication histories of the patients were recorded, including whether angiotensin-converting enzyme inhibitors (ACEIs) or angiotensin II receptor blockers (ARBs) and immunosuppressants were used at or prior to biopsy. Body mass index (BMI) was obtained by dividing the weight of the patient by the square of their height. Mean arterial pressure (MAP) was calculated as the sum of the patient’s systolic blood pressure and twice the diastolic blood pressure divided by three. The estimated glomerular filtration rate (eGFR) was calculated according to the equations developed by George J. Schwartz et al. [17]:eGFR (ml/min/1.73 m^2^) = 39.1[height (m)/Scr (mg/dL)]^0.516^ [1.8/cystatin C (mg/L)]^0.294^ [30/BUN (mg/dL)]^0.169^[1.099] *^male^* [height (m)/1.4]^0.188^
where the value is 1 if the patient is male and 0 if the patient is female. Calculations can be made by bringing in each value according to the corresponding formula unit. Formulations for estimating GFR using endogenous biochemical markers are very useful in clinical medicine because they can be calculated to derive the patient’s renal function at each visit, thus obtaining a consistent and continuously comparable GFR and further determining the patient’s condition and the drug efficacy.

These children were divided into mild, moderate, and severe groups according to the classification criteria for the severity of IgAVN in the SHARE initiative [6]. All the children underwent a percutaneous renal biopsy within 2 days after MRI examination. Under the guidance of an ultrasound, the nephrologist punctured the lower pole of the patient’s right kidney and took biopsy specimens of at least eight glomeruli. All the specimens were sent for immunofluorescence optical microscopy and evaluated for Oxford classification by a professional pathologist with 18 years of experience who was blinded to the information of the children.

### 2.3. The International IgA Nephropathy Prediction Tool in Children

The International IgA Nephropathy Prediction Tool in children uses the available clinical factors and the MEST-C scores to quantify the prognostic risk in children with IgAN and works for children of different races all over the world [18]. The prognostic risk in children with IgAVN was also calculated using this tool. The 80-month risk score of each patient, which represents the risk of a 30% decline in eGFR or ESKD within 80 months after renal biopsy, was calculated using QxMD (https://qxmd.com/calculate/calculator_713/international-igan-prediction-tool-at-biopsy-pediatrics, accessed on 5 January 2024). Children with 80-month risk scores > 10% were categorized into the high-risk group, while the others were placed into the low-risk group.

### 2.4. MRI Techniques

All children who underwent MRI examination were asked to fast for 8 h and abstain from drinking water for 4 h prior to the examination. They followed all the safety precautions and the breathing instructions as trained by us.

A 3-Tesla scanner (MAGNETOM Skyra, Siemens Healthineers, Erlangen, Germany) equipped with an 18-channel phased-array coil was used for the renal imaging. Conventional axial T1-weighted images were taken first, followed by axial and coronal T2-weighted images. The axial BOLD-MRI applied multiple gradient echo sequences and the following parameters: field of view (FOV), 380 × 285 mm; slice thickness, 5.0 mm; matrix, 192 × 154; TR, 179 ms; echoes, 7 equally spaced (2.46–17.22 ms); averages, 1. The images were collected during breath-holding at the end of exhalation. This process was repeated five times, resulting in a total acquisition time of 45–50 s. The ZOOMit (version N, Siemens Healthcare) DWI provides a better image quality and apparent diffusion coefficient (ADC) reproducibility for renal imaging through a reduced FOV and dynamic, spatially selective radiofrequency excitation pulse techniques [19]. Therefore, it is more suitable for evaluating renal function and renal fibrosis than the conventional DWI [20]. The axial ZOOMit DWI used a single-shot echoplanar imaging sequence by excitation of the zoomed FOV and fat-saturation technology to reduce the chemical shift artifacts. The parameters were as follows: FOV, 288 × 125 mm; slice thickness, 4.0 mm; matrix, 120 × 120; TR, 7700 ms; TE, 71 ms. Ten *b*-values (0, 20, 50, 80, 100, 200, 500, 800, and 1000) were used, with a parallel imaging factor of 2 and an acquisition time of 5 min 30 s

### 2.5. Image Analysis and Data Measurement

BOLD-MRI. The BOLD-MRI images were imported to a post-processing workstation (version, VE40B; Siemens Healthineers). Two radiologists (Zhouyan Liao and Guanjie Yuan), without knowing the patient information, selected the slice at the largest level of the kidney hilum to manually draw the regions of interest (ROIs). A 4–5 cm^2^ ringlike cortical ROI and three circular medullary ROIs (1–2 cm^2^) were outlined on the T2* pseudo-color maps, excluding the vessels, renal columns, and cysts. Further details were described by Liang et al. [13]. The average T2* values of the ROIs were automatically calculated by the workstation.

ZOOMit DWI. The ZOOMit DWI images with *b*-values = 0 and 800 s/mm^2^ were first transferred to the MR Body Diffusion Toolbox (v1.5.0, Siemens Healthcare) software. The ROIs of the cortex and the medulla were delineated using the same method mentioned above, using axial T2-weighted images as reference standards. The average value of the ADC was automatically calculated by the software according to the monoexponential model:S_b_/S_0_ = e^−bADCmon^
where S_0_ is the signal intensity for *b* = 0 s/mm^2^, and S_b_ is the signal intensity for *b* = 800 s/mm^2^.

Cortical thickness. The renal cortical thickness of each patient was measured on the slice at the largest level of the kidney hilum of the axial T1 images.

The cortical thickness, T2* values, and ADC values for each patient were calculated as the mean value of the bilateral kidneys. The respective MRI results are shown in Figure 1, and the MRI parameters are shown in Figure 2. Figure 2 shows the MRI parameters and the cortical thickness of the high-risk and low-risk groups.

### 2.6. Statistical Analysis

SPSS (version 26.0, IBM SPSS, Chicago, IL, USA) and GraphPad Prism (version 9.5.1, GraphPad Software Inc., San Diego, CA, USA) were used for the statistical analysis. The two-tailed significance level of *p* < 0.05 was applied. The Shapiro–Wilk test was used to assess the normality of the continuous variables, which was shown as mean ± standard deviation. The categorical data were compared using the chi-square test and displayed in the form of numerical values and percentages. A univariate logistic regression analysis was applied to assess the association between these clinical and MRI parameters and the high-risk group. All the clinical and MRI parameters with *p* < 0.05 were further incorporated into multivariate stepwise logistic regression analyses. Odds ratios (ORs) and 95% confidence intervals (CIs) were also recorded. A preliminary logistic regression model for predicting the high-risk group was based on the results. Receiver operating characteristic (ROC) curves were drawn and the areas under the curve (AUCs) were calculated to evaluate the diagnostic performance of the meaningful variables in a multivariate regression analysis and their combination model for differentiating the high-risk group from the low-risk group.

In addition, the Spearman’s rank correlation was applied to assess the relationships between the MRI parameters and the MEST-C scores. A correlation coefficient above 0.7 was considered strong, between 0.4 and 0.7 was considered moderate, and below 0.4 was considered low. ROC analyses with AUCs were used to evaluate the diagnostic performance of the MRI parameters for differentiating M0 from M1, S0 from S1, and T0 from T1/2.

## 3. Results

### 3.1. Clinical and Pathological Characteristics

Forty-four children (34 males and 10 females; mean age, 9.84 years; age range 5–17 years) who were pathologically confirmed as having IgAN or IgAVN were included. There were 25 children with an 80-month risk score ≤ 10% and 19 children with and 80-month risk score > 10%. The clinical information for the 44 included children is shown in Table 1. As for pathological characteristics, there were significant differences in the M, S, and T scores between patients in the high- and low-risk groups; but not in the E and C scores (Table 2). 

### 3.2. Comparisons between 80-Month Risk Scores ≤10% and 80-Month Risk Scores >10%

The univariate logistic regression analysis showed that the eGFR (*p* = 0.044), 24 h-Upro (*p* = 0.04), MAP (*p* = 0.001), SHARE initiative classification (*p* = 0.015), ADC_Cortex_ (*p* = 0.003), ADC_Medulla_ (*p* = 0.005), and T2*_Cortex_ (*p* < 0.001) were correlated with high-risk children. In the multivariate model, T2*_Cortex_ (OR, 0.512; 95% CI [0.327–0.801]; *p* = 0.003) and MAP (OR, 1.190; 95% CI [1.304–1.370]; *p* = 0.016) were independent significant predictors in high-risk patients with IgAN or IgAVN (Table 3). We produced a nomogram of the preliminary prognostic risk model (Figure 3). As shown in Figure 4, the AUCs of the T2*_Cortex_ and MAP as independent significant predictors were 0.907 (95% CI [0.812–1]) and 0.881 (95% CI [0.780–0.983]), respectively. The AUC of the preliminary model of the T2*_Cortex_ and MAP was 0.947 (95% CI [0.868–1]).

### 3.3. Correlations of MRI Parameters and MEST-C Scores

T2*_Cortex_ and ADC_Medulla_ were both moderately negatively correlated with the S scores (r = −0.419, *p* = 0.005; r = −0.423, *p* < 0.001, respectively) and weakly negatively correlated with the M (r = −0.313, *p* = 0.039; r = −0.396, *p* = 0.008, respectively) and T scores (r = −0.307, *p* = 0.042; r = −0.300, *p* = 0.047, respectively). ADC_Cortex_ was moderately negatively correlated with the M (r = −0.501, *p* = 0.002) and T scores (r = −0.409, *p* = 0.006), while all the MRI parameters used had no correlation with the E or C scores (Figure 5).

As shown in Figure 6, ADC_Cortex_ showed the largest AUC for differentiating M0 from M1 (AUC, 0.921; 95% CI [0.810–1]) and T0 from T1/T2 (AUC, 0.872; 95% CI [0.741–1]). ADCMedulla generated the highest AUC of 0.791 (95% CI [0.633–0.949]) for differentiating S0 from S1. In addition, ADCMedulla had a high AUC for differentiating M0 from M1 (AUC, 0.833; 95% CI [0.611–1]) and differentiating T0 from T1/T2 (AUC, 0.774; 95% CI [0.521–1]). The AUC values of T2*_Cortex_ as an independent significant predictor for the M, S, and T scores were 0.763 (95% CI [0.587–0.939]), 0.788 (95% CI [0.624–0.953]), and 0.779 (95% CI [0.578–0.981]), respectively.

## 4. Discussion

In this study, we constructed a preliminary prognostic risk model of children with IgAN or IgAVN based on clinical and MRI parameters. We found that T2*_Cortex_ and MAP could be potential biomarkers differentiating the high-risk group from the low-risk group in children with IgAN or IgAVN. The preliminary prognostic risk model of T2*_Cortex_ and MAP was the most valuable (AUC 0.947) in predicting the high-risk group.

This study showed that the SHARE initiative classification, eGFR, and 24 h-Upro were significant in the univariate logistic regression analysis but not in the multivariate analysis. The SHARE initiative classification judges the severity of IgAVN based on eGFR and PCR, which provides a simple and convenient basis for clinicians to make subsequent diagnostic and therapeutic care decisions. Estimated eGFR is one of the most commonly used parameters in the clinical diagnosis and treatment of CKD, and its decline reflects impaired renal function and the progression of chronic kidney disease [21]. Although the mean eGFR values were lower in children in the high-risk group, they were still within the normal range. Pediatric IgAN and IgAVN are generally considered benign because very few patients reach ESKD before adulthood [22]. However, renal damage continues to develop even decades after the initial symptoms disappear [23]. Therefore, baseline eGFR is not a good predictor of prognosis.

PCR and 24 h-Upro are consistent in evaluating proteinuria in children with CKD, such as IgAN and IgAVN [24]. Urinary protein is considered to be a clinically important predictor of the long-term renal prognosis of IgAN or IgAVN [25]. Reducing 24 h-Upro and PCR protects the kidneys and delays disease progression, which is supported by numerous studies [25,26,27]. In the early stage of CKD, the molecular barrier and charge barrier of the glomerular filtration membrane are disrupted, resulting in increased urinary albumin [28]. As CKD progresses, glomerular permeability continues to deteriorate and renal tubular reabsorption and the metabolism of urinary protein decrease, leading to a continued increase in the total protein in urine [28]. However, in this study, we found that although the 24 h-Upro and SHARE initiative classification had a certain predictive effect, they were not independent predictors in the high-risk group. A possible reason is that our study included the T2* value as an independent predictor, which is a more powerful predictor than the 24 h-Upro and SHARE initiative classification. Previous studies have shown that the T2* value is significantly correlated with proteinuria in patients with CKD [29], and that the T2* value is an independent predictor in patients with CKD regardless of the degree of proteinuria and eGFR [30].

Among the many clinical parameters, only MAP was a valuable independent parameter after excluding the effects of urinary protein, eGFR, and other factors for predicting the high-risk group, which was similar to the research of Dionne et al. [31]. Hypertension is an important risk factor in the incidence and progression of CKD. It increases intraglomerular cystic pressure, leading to glomerular fibrosis, atrophy, and renal arteriosclerosis, resulting in renal parenchyma ischemia and glomerulopenia [32,33]. In addition, hypertension is significantly more prevalent in patients with chronic kidney disease. This is because the loss of nephrons leads to water and sodium retention, which results in an increased extracellular volume, activation of the renin–angiotensin–aldosterone system, and decreased sodium excretion [34]. Many guidelines, including the American Academy of Pediatrics Clinical Practice Guidelines and the KDIGO guidelines, recommend that children with CKD need to be monitored and their blood pressure controlled closely to protect renal function and slow the progression of CKD [35]. All of these points suggest that CKD and hypertension have a mutually reinforcing effect. Therefore, controlling the blood pressure is very important in patients with CKD. ACEIs or ARBs are recommended as a first-line medication, because they are not only effective in controlling blood pressure, but also reduce urinary protein [36].

The above points suggest that the commonly used clinical indicators have a limited predictive ability in the prognosis of pediatric patients with IgAN or IgAVN. In addition, the International IgA Nephropathy Prediction Tool contains many other variables besides MAP, suggesting that MAP alone may not be as predictive as is ideal. Obviously, it is not sufficient to use clinical indicators to build an ideal non-invasive prognostic risk model for children with IgAN or IgAVN, so we added functional MRI parameters to co-construct the model. The T2* values were greater in the cortex and medulla of the kidneys in the low-risk group, suggesting that renal oxygenation was lower in the high-risk group. The ADC values were also lower in the high-risk group than in the low-risk group, suggesting that the diffusion of water molecules in the kidney is more restricted in the high-risk group. These results suggest that there are more renal microstructural and pathophysiological changes in high-risk individuals. A functional MRI can provide an early indication of these renal microstructural changes and thus help in the early non-invasive identification of high-risk individuals.

The BOLD-MRI has been widely used to assess the kidney oxygenation status in vivo. As the oxygen in the blood content directly affects the level of deoxyhemoglobin, it increases the magnetic spin dephasing of the water protons, which leads to a decrease in the T2* values. It has the advantages of being non-invasive and contrast-free, making it suitable for the long-term follow-up of patients with CKD. Here, we found that the T2*_Cortex_ was a significant independent predictor for the high-risk group of children with IgAN or IgAVN. In this study, the AUC values of the T2*_Cortex_ were significantly high, which is consistent with the studies of Liang et al. [13] and Zhou et al. [37]. The T2* values of the kidney cortex were greater in the low-risk group, indicating that renal cortical oxygenation was lower in the high-risk group. Most of the blood supply to the kidney went to the cortex. Therefore, when the partial pressure of oxygen in the renal cortex was low, it meant that the entire kidney was in a state of hypoxia. Hypoxia induces the infiltration and activation of inflammatory cells [38], renal tubular cell injury, endothelial cell dysfunction and apoptosis, and the differentiation into myofibroblasts [39]. These mechanisms lead to inflammation, the loss of renal capillary structure, and renal fibrosis [35,39], which in turn exacerbates renal hypoxia, forming a vicious cycle. Therefore, measuring the oxygenation of the kidney tissue will help clinicians to appropriately increase treatments to increase renal oxygenation and nephroprotective effects in children with IgAN or IgAVN, especially in the early stages of the diseases, thereby delaying disease progression and improving the prognosis. However, unlike T2*_Cortex_, T2*_Medulla_ was not a significant risk factor for the high-risk group in this study, which is the same as in the study of Zhou et al. [37] but different from the study of Liang et al. [13]. A possible reason is that the research population in Liang et al. comprised adults with IgAN, whereas ours comprised children with IgAN or IgAVN. More importantly, the eGRF, MAP, and urine protein values differed significantly between the high-risk and low-risk groups in the study by Liang et al. However, they only conducted a univariate analysis and did not perform a further multivariate analysis to rule out other clinical factors. It has been reported that the outer medulla is extremely sensitive to vascular congestion and that the induction of tachycardia leads to a significant decrease in the medullary R2* (1/T2*). Therefore, we believe that medullary oxygenation is susceptible to a very large number of factors and should be studied to exclude as many confounding factors as possible. In Zhou’s study, a BOLD-MRI was performed before and after the injection of furosemide to assess the results, which we believe provides more convincing results for T2*_Medulla_.

ZOOMit DWI. ADC_Cortex_ and ADC_Medulla_ were both significant in the univariate logistic regression analyses, but not in the multivariate analyses. The ADC values of the kidney were lower in the high-risk group, which is consistent with the study of Dillman et al. [20]. In addition, ADC_Cortex_ and ADC_Medulla_ were correlated with the M, S, and T scores and had a high diagnostic efficiency for the M, S, and T scores. Furthermore, there were significant differences in the M, S, and T scores between children in the high- and low-risk groups, but not in the E and C scores. These results suggest that the ADC values reflect pathological changes, such as the degree of interstitial fibrosis, and other renal microstructural changes, including the Oxford classification [15]. Children with lower ADC values indicated more renal microstructural changes, requiring a more targeted treatment regimen to minimize adverse outcomes. However, the ability of the ADC values to predict the prognostic risk of the children was limited. This may be related to the complexity of the International IgA Nephropathy Prediction Tool [18], where prognosis is not entirely dependent on changes in the kidney microstructure.

We will continue to collect data on an ongoing basis and add further data in the future to validate our conclusions. We will also conduct additional MRI studies on renal function in children, including an investigation of the predictive values of various MRI sequences in the prognosis of patients with reduced renal function to find the optimal sequence. We thus aim to construct practical and reliable non-invasive models to provide guidance for clinical work and help early identification of high-risk patients for early intervention and treatment. In addition, in the future, our studies are expected to further update the clinically recognized prediction formulas, including the International IgA Nephropathy Prediction Tool, so that the histopathological parameters therein can be replaced with functional MRI parameters.

This study has the following limitations: first, this was a single-center study with a relatively small number of children with IgAN or IgAVN; second, the prognostic risk stratification scores were obtained only using the International IgA Nephropathy Prediction Tool and do not represent the true prognostic outcomes of the patients; and third, respiration, bowel movements, and large vessel pulsations can easily affect the image quality of the BOLD-MRI. Some younger children could not hold their breath very well, even though they had received our breathing training and used abdominal belts during the MRI examination.

## 5. Conclusions

The SHARE initiative classification, 24 h-Upro, and other clinical indicators have limited the ability to predict the prognosis of children with IgAN or IgAVN. Our prognostic model, combining functional MRI parameters and clinical indicators (T2*_Cortex_ and MAP), plays an important role in the non-invasive prognosis of children with IgAN or IgAVN.

## Figures and Tables

**Figure 1 bioengineering-11-00750-f001:**
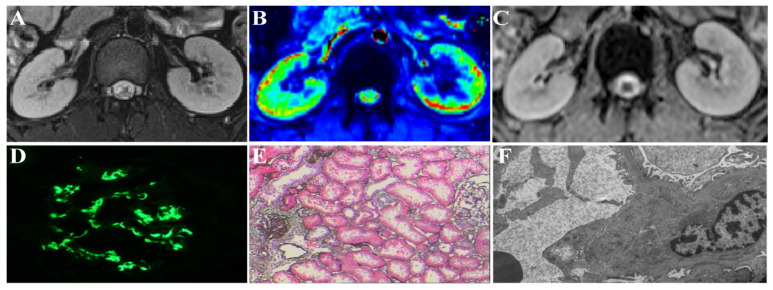
Respective MRI images and histology images of a patient with IgAN (16 years old, male). (**A**–**F**) Axial T2-weighted image, T2* pseudo-color image, ADC image, immunofluorescence map, light microscopy map, and electron microscopy map, respectively.

**Figure 2 bioengineering-11-00750-f002:**
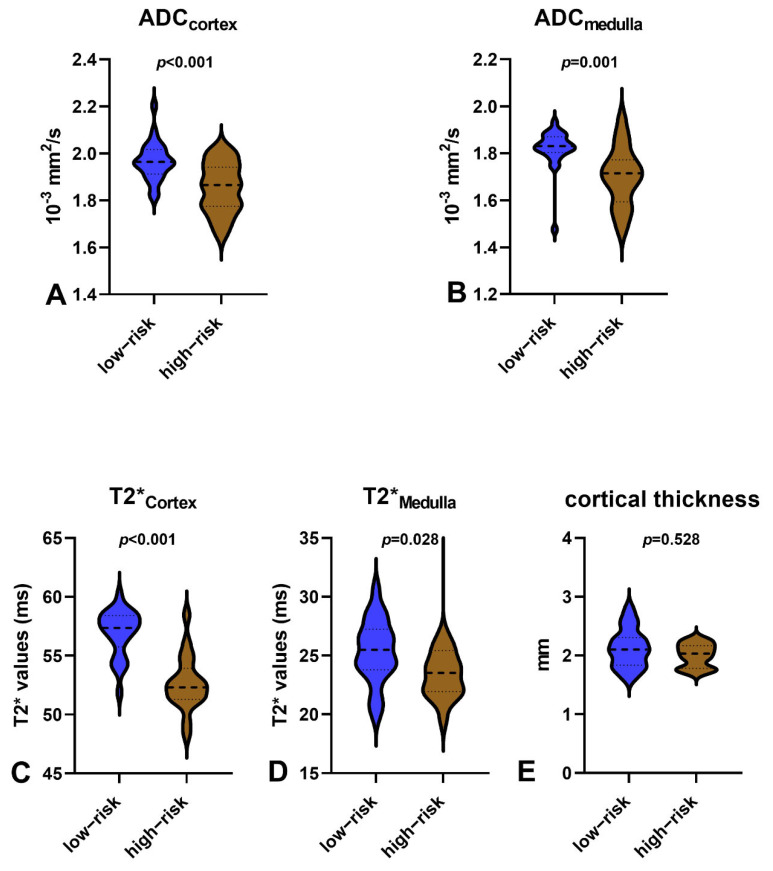
MRI parameters between children with an 80-month risk score ≤10% and children with an 80-month risk score >10%. (**A**) ADC_Cortex_ values; (**B**) ADC_Medulla_ values; (**C**) T2*_Cortex_ values; (**D**) T2*_Medulla_ values; (**E**) cortical thickness.

**Figure 3 bioengineering-11-00750-f003:**
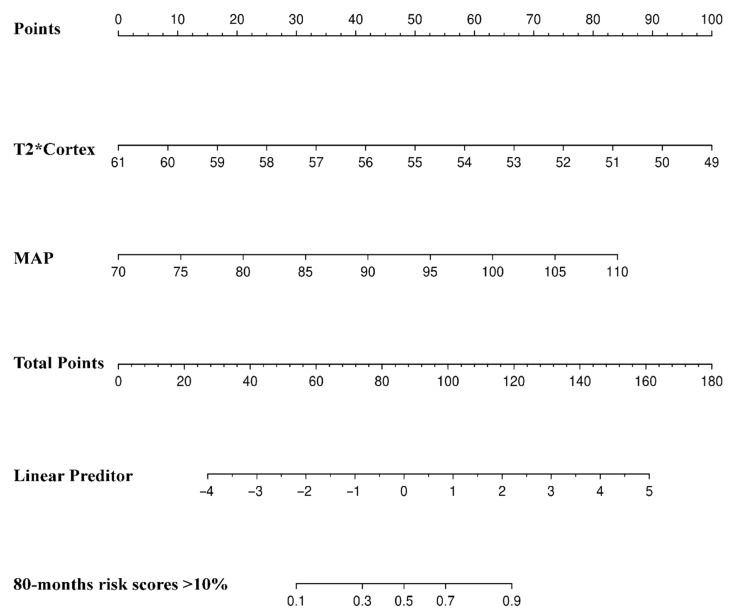
Nomogram for predicting the probability of the high-risk (80-month risk scores >10%) group of children with IgAN or IgAVN. Points were assigned for the T2*_Cortex_ values and mean arterial pressure (MAP) by drawing a line upward from the corresponding values to the “Points” line. “Total Points” are calculated as the sum of the individual score of the T2*_Cortex_ and MAP.

**Figure 4 bioengineering-11-00750-f004:**
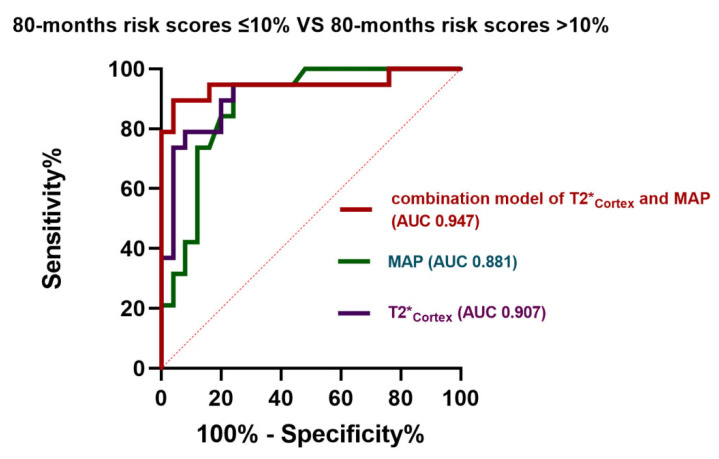
ROC curves for the diagnostic performance of the T2*_Cortex_ values, the mean arterial pressure (MAP), and the combination model of the T2*_Cortex_ values and MAP for differentiating the 80-month score ≤10% group and the 80-month risk score >10% group.

**Figure 5 bioengineering-11-00750-f005:**
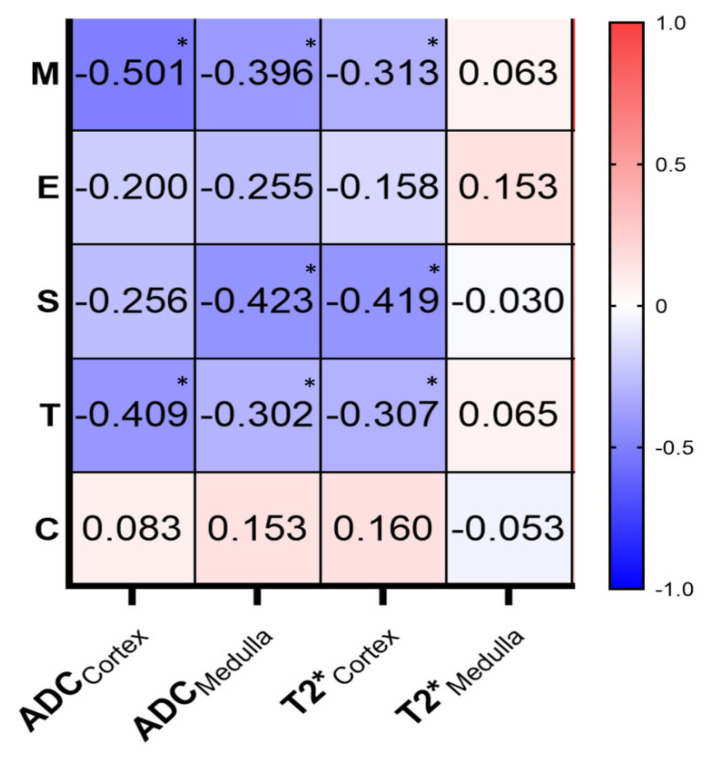
Correlations of the MRI parameters and the pathological data (MEST-C scores). * Significant result.

**Figure 6 bioengineering-11-00750-f006:**
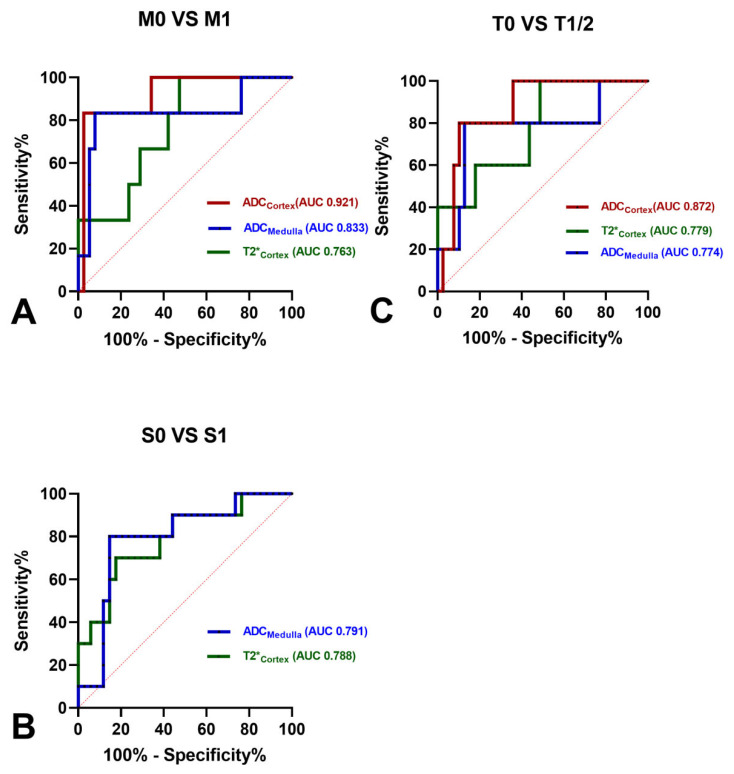
ROC curves for the diagnostic performance of the MRI parameters for differentiating the (**A**) M0 from M1, (**B**) S0 from S1, and (**C**) T0 from T1/2.

**Table 1 bioengineering-11-00750-t001:** Baseline information on the patients.

Variables	All	Low-Risk	High-Risk
	*n* = 44	*n* = 25	*n* = 19
sex			
Male	34 (77.27)	19 (76.00)	15 (78.95)
Female	10 (22.73)	6 (3.00)	4 (21.05)
age (y)	9.84 ± 2.74	9.28 ± 2.46	10.58 ± 2.97
Height (cm)	144.27 ± 16.26	140.60 ± 15.52	149.11 ± 16.35
Weight (kg)	38.62 ± 15.40	35.83 ± 11.25	42.28 ± 13.17
BMI (kg/m^2^)	18.15 ± 3.37	17.76 ± 3.09	18.67 ± 3.73
eGFR (ml(min × 1.73 m^2^))	118.49 ± 29.16	126.55 ± 30.83	107.89 ± 23.56
Urea (mmol/L)	5.37 ± 2.45	5.35 ± 3.04	5.40 ± 1.41
Scr (μmol/L)	48.16 ± 17.27	44.2 ± 17.74	53.37 ± 15.57
Uric acid (μmol/L)	287.44 ± 91.38	292.24 ± 90.71	281.26 ± 94.37
HCO_3_− (mmol/L)	22.73 ± 2.38	22.29 ± 2.36	23.30 ± 2.36
NLR	2.40 ± 2.01	1.92 ± 1.33	3.01 ± 2.57
β2-MG (mg/L)	0.55 ± 0.76	0.33 ± 0.33	0.84 ± 1.04
cystantin C (mg/L)	0.96 ± 0.27	0.91 ± 0.22	1.02 ± 0.32
24 h-Upro (g)	1.15 ± 1.56	0.64 ± 0.98	1.83 ± 1.91
24 h-PCR (mg/mmol)	219.00 ± 367.00	128.70 ± 184.78	337.80 ± 500.00
Urinary occult blood			
1+	5 (11.36)	3 (12.00)	2 (10.53)
2+	10 (22.73)	9 (36.00)	1 (5.26)
3+	29 (65.91)	13 (52.00)	16 (84.21)
SBP (mmHg)	107.25 ± 14.21	99.76 ± 9.75	117.11 ± 13.24
DBP (mmHg)	74.98 ± 10.48	70.16 ± 7.73	81.32 ± 10.40
MAP (mmHg)	85.73 ± 11.00	80.03 ± 7.72	93.25 ± 10.21
The SHARE initiative			
Mild	34 (77.27)	23 (92.00)	11 (57.89)
Moderate/Severe	10 (22.73)	2 (8.00)	8 (42.11)

BMI, body mass index; eGFR, estimated glomerular filtration rate; Scr, serum creatinine; NLR, neutrophil-to-lymphocyte ratio; β2-MG, urinary β2-microglobulin; 24 h-Upro, 24 h-urinary protein; 24 h-PCR, 24 h-urinary protein/creatinine ratio; SBP, systolic blood pressure; DBP, diastolic blood pressure; MAP, mean arterial pressure.

**Table 2 bioengineering-11-00750-t002:** Oxford classification of patients.

Variable			All	Low-Risk	High-Risk	*p*-Value
			*n* = 44	*n* = 25	*n* = 19	
M	M0	Absent to ≤50% of glomeruli	38	25	13	0.002 *
	M1	>50% of glomeruli	6	0	6	
E	E0	Absent	36	21	15	0.667
	E1	Present	8	4	4	
S	S0	Absent	34	24	10	0.001 *
	S1	Present	10	1	9	
T	T0	Absent to ≤25% of the cortex	39	25	14	0.006 *
	T1/2	25–50% of the cortex or>50% of the cortex	5	0	5	
C	C0	Absent	17	9	8	0.68
	C1/2	1%–24% of the glomeruli or≥25% of the glomeruli	27	16	11	

M, mesangial hypercellularity; E, endothelial hypercellularity; S, segmental sclerosis or adhesion; T, interstitial fibrosis or tubular atrophy and C, crescent. * Significant result.

**Table 3 bioengineering-11-00750-t003:** Univariate and multivariate logistic regression analysis for predicting 80-month risk scores >10%.

Variable	Univariate Analysis			Multivariate Analysis		
	OR	95%CI	*p*-Value	OR	95%CI	*p*-Value
sex						
male	0.844	0.201–3.546	0.817			
female						
Age (y)	1.203	0.95–1.525	0.126			
BMI (kg/m^2^)	1.806	0.906–1.302	0.373			
eGFR(ml(min × 1.73 m^2^))	0.975	0.951–0.999	0.044 *			0.965
Urea (mmol/L)	1.009	0.789–1.289	0.946			
Scr (μmol/L)	1.035	0.994–1.077	0.096			
Uric acid (μmol/L)	0.999	0.992–1.005	0.687			
HCO_3_− (mmol/L)	1.212	0.919–1.598	0.173			
NLR	1.354	0.951–1.929	0.093			
β2-MG (mg/L)	6.039	0.891–40.929	0.066			
cystantin C (mg/L)	5.364	0.747–60.646	0.175			
24 h-Upro (g)	1.001	1–1.001	0.04 *			0.675
24 h-PCR (mg/mmol)	1.003	0.999–1.006	0.116			
Urinary occult blood	2.123	0.809–6.052	0.122			
MAP (mmHg)	1.239	1.089–1.409	0.001 *	1.190	1.304–1.370	0.016 *
The SHARE initiative						
Mild	8.364	1.516–46.148	0.015 *			0.367
Moderate/Severe						
MRI parameters						
ADC_Cortex_	0.986	0.977–0.995	0.003 *			0.503
ADC_Medulla_	0.989	0.981–0.997	0.005 *			0.969
T2*_Cortex_	0.457	0.302–0.693	<0.001*	0.512	0.327–0.801	0.003*
T2*_Medulla_	1.007	0.987–1.028	0.479			
cortical thickness (mm)	3.598	0.397–32.627	0.255			

BMI, body mass index; eGFR, estimated glomerular filtration rate; Scr, serum creatinine; NLR, neutrophil-to-lymphocyte ratio; β2-MG, urinary β2-microglobulin; 24 h-Upro, 24 h-urinary protein; 24 h-PCR, 24 h-urinary protein/creatinine ratio; MAP, mean arterial pressure; ADC, apparent diffusion coefficient. * Significant result.

## Data Availability

The datasets used and/or analyzed in the current study are available from the corresponding author on reasonable request.

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
