# Peer review of "Explore the Value of Multi-Parameter MRI in Non-Invasive Assessment of Prognostic Risk and Oxford Classification in Children with IgAN or IgAVN"

_bioengineering, 2024, doi:10.3390/bioengineering11080750_

Round 1
Reviewer 1 Report
Comments and Suggestions for Authors
This manuscript reports the application of a multi-parameter approach to interpreting MR images to define prognostic risk and perform Oxford classification for children with IgAN or IgAVN. The manuscript lays out a reasonable approach to projecting prognostic risk and performing these classifications. The authors appropriately acknowledge the small size of the sample set but as a preliminary finding, the use of T2* has promise as a prognostic parameter for use with chronic kidney diseases and especially in use with children. The figures and tables adequately summarise the findings and the discussion text is an appropriate and balanced interpretation of the data presented. There are some points that the authors could address and these are detailed below.
1. Whilst the content of the manuscript is fairly comprehensible, the manuscript does have a number of grammatical and syntactic errors that should be corrected (too many to list individually).
2. Page 2 lines 85-86 “GFR . . . 1.4]0.188”. Whilst this equation is clearly derived from the content in reference 13, it is still not clear exactly how the derivation has been made as it does not appear to be an exact replication of the computations performed in reference 13. Also, the word ‘male’ has been included in the formula in superscript and it is not clear what you are meaning to say here. Can you include in the manuscript a more detailed explanation of the use and meaning of this formula?
3. Page 6 lines 189-191 “Univariate logistic regression . . . with high-risk children.” I think that these important observations should be discussed in a bit more detail. As indicated in Figure 2, what you are seeing for a number of these variables is that the values are lower for the high-risk group compared to the low-risk group. I think this deserves some discussion and some suggestions on why this might be so.
4. Page 11 lines 288-289 “We found the . . . or IgAVN.” Independent of what? If it is an independent predictor then you should clarify here under which sets of conditions and in relation to which parameters it remains an independent predictor.
5. Page 12 lines 302-306 “However, unlike . . . IgAN or IgAVN.” I think this important point merits some more discussion here. If your assertion is correct, then why should your study differ from the Liang study on the basis of the age of the subjects and yet did not differ from the Zhou study, which was not designed to explicitly study children? I think a more consistent and detailed consideration of the possible factors involved here would be useful.
Comments on the Quality of English LanguageThe manuscript needs to be revised to correct grammatical and syntactic errors.
Author Response
Comments: This manuscript reports the application of a multi-parameter approach to interpreting MR images to define prognostic risk and perform Oxford classification for children with IgAN or IgAVN. The manuscript lays out a reasonable approach to projecting prognostic risk and performing these classifications. The authors appropriately acknowledge the small size of the sample set but as a preliminary finding, the use of T2* has promise as a prognostic parameter for use with chronic kidney diseases and especially in use with children. The figures and tables adequately summarized the findings and the discussion text is an appropriate and balanced interpretation of the data presented. There are some points that the authors could address and these are detailed below.
Response:Thank you very much for your valuable comments and affirmation of our article!
Comments 1: Whilst the content of the manuscript is fairly comprehensible, the manuscript does have a number of grammatical and syntactic errors that should be corrected (too many to list individually)
Response 1: Thank you very much for your comments and we apologized for any reading discomfort. We had a professional English editing for our articles.
Comments 2: Page 2 lines 85-86 “GFR . . . 1.4]0.188”. Whilst this equation is clearly derived from the content in reference 13, it is still not clear exactly how the derivation has been made as it does not appear to be an exact replication of the computations performed in reference 13. Also, the word ‘male’ has been included in the formula in superscript and it is not clear what you are meaning to say here. Can you include in the manuscript a more detailed explanation of the use and meaning of this formula?
Response 2: Thank you very much for your comments. We used the exact formula in the abstract in reference 13, which was the formula derived in the text for the optimal calculation of eGFR in children. This formula was established by the authors using baseline visit data from 349 children from the pediatric CKD cohort. The authors of this article, who have been involved in studies of kidney function, synthesized and compared other previous formulas and incorporated factors previously thought to have little relationship with eGFR. After performing linear regression analyses to assess precision, goodness-of-fit, and accuracy, the improvement resulted in a formula for estimating GFR based on height, serum creatinine, cystatin C, blood urea nitrogen, and sex, which has superior performance to other formulas and is widely recognized and used internationally. Where male means that if the patient is male, the value here is 1; if the patient is female, the value here is 0. Calculations can be made by bringing in each value according to the corresponding formula unit. Formulations for estimating GFR using endogenous biochemical markers are very useful in clinical medicine because they can be calculated to derive the patient's renal function at each visit, thus obtaining a consistent and continuously comparable eGFR and further determining the patient's condition and drug efficacy. It is not necessary or practical for patients to undergo insulin removal test or single-photon emission computed tomography at each visit. Whereas the physiological states of children and adults differ very significantly, previous studies have also demonstrated that adult formulas cannot be used to estimate eGFR values in children. In contrast, the formula used in this paper used readily available biochemical markers to estimate GFR and has been shown to be accurate and easy to promote for clinical use.
Comments 3: Page 6 lines 189-191 “Univariate logistic regression . . . with high-risk children.” I think that these important observations should be discussed in a bit more detail. As indicated in Figure 2, what you are seeing for a number of these variables is that the values are lower for the high-risk group compared to the low-risk group. I think this deserves some discussion and some suggestions on why this might be so.
Response 3: Thank you very much for your comments. We have added the discussion of the univariate logistic regression results in the discussion part.
Comments 4: Page 11 lines 288-289 “We found the . . . or IgAVN.” Independent of what? If it is an independent predictor then you should clarify here under which sets of conditions and in relation to which parameters it remains an independent predictor.
Response 4: Thank you very much for your comments. We are very sorry for the ambiguity of the sentence here. Because MAP was still statistically significant in multivariate logistic regression analyses, this suggested that it is still significant in our study after excluding the effects of urinary protein, eGFR and other factors. Considering the inconvenience this brings to understanding, we also removed the word independent.
Comments 5: Page 12 lines 302-306 “However, unlike . . . IgAN or IgAVN.” I think this important point merits some more discussion here. If your assertion is correct, then why should your study differ from the Liang study on the basis of the age of the subjects and yet did not differ from the Zhou study, which was not designed to explicitly study children? I think a more consistent and detailed consideration of the possible factors involved here would be useful.
Response 5: Thank you very much for your comments. eGFR, MAP, and urine protein values differed significantly between the high-risk and low-risk groups in the study by Liang et al. But they only did a univariate analysis and did not do further multivariate analysis to rule out other clinical factors. It has been reported that the outer medulla is extremely sensitive to vascular congestion and that injection of tachycardia leads to a significant decrease in medullary R2* (1/T2*). Therefore, we believed that medullary oxygenation is susceptible to a very large number of factors and should be studied to the exclusion of as many other factors as possible. In Zhou's study, BOLD-MRI was performed before and after the injection of furosemide to assess the results, so we thought it's a more convincing results of T2*Medulla.

Reviewer 2 Report
Comments and Suggestions for Authors
1. In the first sentences of the Introduction section, you can add some statistics about the prevalence or mortality rate of children with IgAN or IgAVN to highlight the importance and necessity of your project for the audience.
2. The last paragraph of the Introduction section should be corrected. Please clarify the hypothesis and the parameters that you want to use to check the correctness of your hypothesis clearly.
3. The literature review is severely lacking and insufficient. You only evaluated four papers, with one from 2023 and one from 2021 or older. This is not enough. You should report the main findings of all related papers and clearly identify their shortcomings and the gaps that you aim to fill. Currently, you abruptly jump to your goal without providing this necessary context.
4. Can you please explain in more detail your exclusion criteria and the logic behind them?
5. The study's sample size is extremely small (44 patients), which severely limits the statistical power and generalizability of the findings. This is particularly concerning given the variability in clinical presentations and outcomes of IgAN and IgAVN.
6. The predictive model lacks external validation. Relying solely on internal validation within a small cohort does not provide sufficient evidence of the model's utility in broader clinical practice.
7. The statistical analysis appears to be inadequately performed. The multivariate logistic regression model includes variables that were not significant in univariate analysis, which is methodologically unsound. The authors do not provide confidence intervals for several key parameters, which are essential for understanding the precision and reliability of the estimates. Can you please explain technically why you have really good AUC values with is small databse with many non-Significant parameters?
8. The manuscript does not present significant advancements beyond existing literature. The findings regarding the use of T2* values and MAP as biomarkers are not novel and have been reported previously in similar contexts. The study does not sufficiently address gaps in the current understanding of non-invasive prognostic tools for pediatric IgAN and IgAVN.
9. The manuscript contains numerous grammatical errors and awkward phrasing, which hinder the clarity and readability of the text. Figures and tables are poorly labeled and do not effectively communicate the study’s findings.
10. The study mentions adherence to the Declaration of Helsinki but does not provide adequate details on ethical approvals and patient consent processes, especially for a pediatric population.
Comments on the Quality of English LanguageLow level
Author Response
Comments 1: In the first sentences of the Introduction section, you can add some statistics about the prevalence or mortality rate of children with IgAN or IgAVN to highlight the importance and necessity of your project for the audience.
Response 1: Thank you very much for your comments. We have added the prevalence of children with IgAN or IgAVN. Pediatric IgAN and IgAVN are generally considered benign because very few patients reach ESKD before adulthood. So we didn't add their mortality rate. The importance and necessity of our research is highlighted mainly because of their high prevalence.
Comments 2: The last paragraph of the Introduction section should be corrected. Please clarify the hypothesis and the parameters that you want to use to check the correctness of your hypothesis clearly.
Response 2: Thank you very much for your comments. We've changed the last paragraph.
Comments 3: The literature review is severely lacking and insufficient. You only evaluated four papers, with one from 2023 and one from 2021 or older. This is not enough. You should report the main findings of all related papers and clearly identify their shortcomings and the gaps that you aim to fill. Currently, you abruptly jump to your goal without providing this necessary context.
Response 3: Thank you very much for your comments, we have added this part to the third paragraph of the introduction.
Comments 4: Can you please explain in more detail your exclusion criteria and the logic behind them?
Response 4: Thank you very much for your comments. Eight children could not hold their breath well resulting in poor image quality and 39 patients did not undergo MRI without images. Therefore, these 47 patients were excluded.
Comments 5: The study's sample size is extremely small (44 patients), which severely limits the statistical power and generalizability of the findings. This is particularly concerning given the variability in clinical presentations and outcomes of IgAN and IgAVN.
Response 5: Thank you very much for your comments. This is indeed an unavoidable limitation of our research. Functional MRI of pediatric patients with CKD are really difficult to collect clinically. Not many parents of children are willing to have their children undergo MR examination because of the limited knowledge of renal functional MRI at present. In addition to this, some children do not cooperate well with MR examinations. Some younger children could not hold their breath very well, even though they had received our breathing training and used abdominal belts during the MR examination. There were also children who cannot remain immobile in the supine position, so this results in heavy image motion artifacts that cannot be used. So we've been doing our best to collect functional MRI images of patients with renal function. We will continue to collect data on an ongoing basis and add further data in the future to validate our conclusions and conduct additional MRI studies of renal function in children.
Comments 6: The predictive model lacks external validation. Relying solely on internal validation within a small cohort does not provide sufficient evidence of the model's utility in broader clinical practice.
Response 6: Thank you very much for your comments. This is indeed an unavoidable limitation of our research. We will continue to collect data on an ongoing basis and add further independent external validation datasets in the future to validate our conclusions. Although the degree of proof of the utility of the model in clinical practice is limited, our model remains a strong guide to clinical work. Our model emphasizes the importance of cortical T2* values and MAP. Clinical practice should pay extra attention to pediatric patients with high MAP or low cortical T2* values to prevent them from developing adverse renal outcomes.
Comments 7: The statistical analysis appears to be inadequately performed. The multivariate logistic regression model includes variables that were not significant in univariate analysis, which is methodologically unsound. The authors do not provide confidence intervals for several key parameters, which are essential for understanding the precision and reliability of the estimates. Can you please explain technically why you have really good AUC values with is small databse with many non-Significant parameters?
Response 7: Thank you very much for your comments. We added confidence intervals for these parameters. Table 3 showed the results of univariate and multivariate logistic regression. Only clinical and MRI parameters that were statistically significant (P<0.05) in the univariate analysis were included in the multivariate logistic regression analyses. Because our multivariate regression used a stepwise approach, eliminating variables with insignificant effects.
These clinical factors included in the analysis are usual clinical parameters commonly used to assess the status of patients with CKD. We wanted to include as many clinical parameters as possible in order to find the best predictors of prognosis in pediatric patients. The final inclusion of clinical plus MRI parameters was high, so we adopted an univariate regression analysis and then multivariate regression analysis. In order to build the optimal regression prediction model, we used the stepwise regression method. So in the end the explanatory variables retained in the model are both significant and free of severe multicollinearity. All variables in the regression model are significant for the dependent variable. So even if many variables are eliminated at the end, the predictive power of our model remains strong, which is the strength of the multivariate stepwise regression approach.
Comments 8 : The manuscript does not present significant advancements beyond existing literature. The findings regarding the use of T2* values and MAP as biomarkers are not novel and have been reported previously in similar contexts. The study does not sufficiently address gaps in the current understanding of non-invasive prognostic tools for pediatric IgAN and IgAVN.
Response 8: Thank you very much for your comments. Although our MR technology is not the most novel, our focus in this article is on our clinical point of view, which is the noninvasive prediction of prognosis for children with IgAN or IgAVN. Most of the studies have performed on adults with CKD, whereas functional MRI has rarely been used in pediatric patients with CKD. Moreover, current studies have focused on the correlation between T2* or ADC and renal function in children, and little attention has been paid to the ability of MRI to predict the prognosis of pediatric CKD patients.
Comments 9 : The manuscript contains numerous grammatical errors and awkward phrasing, which hinder the clarity and readability of the text. Figures and tables are poorly labeled and do not effectively communicate the study’s findings.
Response 9: Thank you very much for your comments and we apologized for any reading discomfort. We've had a professional agency touch up our articles. We've double-checked the article and fixed these issues.
Comments 10 : The study mentions adherence to the Declaration of Helsinki but does not provide adequate details on ethical approvals and patient consent processes, especially for a pediatric population.
Response 10: Thank you very much for your comments. This retrospective study adhered to the Declaration of Helsinki's rule of ethics and was approved by the ethic committee of Huazhong University of Science and Technology, Tongji Hospital, Tongji Medical College. The requirement for informed consent was waived.

Reviewer 3 Report
Comments and Suggestions for Authors
This study explored the use of multi-parametric MRI in combination with clinical factors for non-invasively providing risk-stratification for children with IgAN/IgAVN. A study weakness is the small sample size, as well as the fact that it was conducted at single institution.
The paper could benefit from editing of the English.
Abstract: For the term “multi-parameter MRI,” consider “multiparametric MRI.”
Abstract: “plays an important and irreplaceable role in noninvasive prognosis of IgAN or IgAVN children”: I think that further research would be needed before making the claim that this biomarker is irreplaceable.
Introduction: This is a bit brief, and it would be helpful to expand upon the relevant literature.
Methods, Line 70: How many patients were excluded because of poor image quality and how many were excluded because of the lack of MR images?
Methods, Line 74: “cystantin C”: Do you mean Cystatin C?
Methods, Line 96: The MEST-C score is introduced here. Please clarify what it represents, as it is related to the Oxford classification.
Methods, Line 164: “differentiating M0 from M1, S0 from S1, and T0 from T1/2”. Please define these terms or refer to Table 2.
Figure 1: The figure includes both MRI and histology images, not just MRI images.
Figure 2: It would be helpful to show p-values for each of A-E.
Results, Line 220: “ADCMedulla generated the highest AUC of 0.791 (95% CI 0.633–0.949) for differentiating S0 from S1, and the AUC for differentiating M0 from M1 (AUC 0.833, 95% CI 0.611-1) and T0 from T1/T2 (AUC 0.774, 95% CI 0.521-1).” This wording is confusing. While ADCMedulla performed best for distinguishing S0 from S1, the wording makes it seem that ADCmedulla was also the best for the other two categories.
Discussion, Line 307-315: “ADC values of kidney were higher in the high-risk group, which was consistent with the study of Dillman et al. [20].” “Children with lower ADC values indicated more renal microstructural changes, requiring a more targeted treatment regimen to minimize adverse outcomes.” In Figure 2, it seems like the high-risk group had lower ADC values.
Discussion: It would be helpful to list future study directions.
Discussion: Would you say that one advantage of this study is that you can obtain prognostic information using imaging and a clinical factor without having to do a biopsy to get histological information, which in turn are variables that constitute the “international IgAN prediction tool at biopsy – adults” (https://qxmd.com/calculate/calculator_713/international-igan-prediction-tool-at-biopsy-pediatrics)? This could be emphasized more in the Discussion.
Discussion: It seems like the goal of the study was to differentiate between high-risk and low-risk groups. However, it might have also been useful to analyze direct clinical outcomes. How good of a gold standard is the International IgAN prediction tool at biopsy?
Comments on the Quality of English LanguageModerate editing of the English is suggested for grammar and diction.
Author Response
Comments 1: This study explored the use of multi-parametric MRI in combination with clinical factors for non-invasively providing risk-stratification for children with IgAN/IgAVN. A study weakness is the small sample size, as well as the fact that it was conducted at single institution.
Response 1: Thank you very much for your comments. This is indeed an unavoidable limitation of our research. Functional MRI of pediatric patients with CKD are really difficult to collect clinically. We will continue to collect data on an ongoing basis and add further data in the future to validate our conclusions and conduct additional MRI studies of renal function in children.
Comments 2: The paper could benefit from editing of the English.
Response 2: Thank you very much for your comments and we apologized for any reading discomfort. We've had a professional agency touch up our articles.
Comments 3: Abstract: For the term “multi-parameter MRI,” consider “multiparametric MRI.”
Response 3: Thank you very much for your comments. We've made the changes as you suggested.
Comments 4: Abstract: “plays an important and irreplaceable role in noninvasive prognosis of IgAN or IgAVN children”: I think that further research would be needed before making the claim that this biomarker is irreplaceable.
Response 4: Thank you very much for your comments. The word irreplaceable really doesn't quite fit, and we've removed this word.
Comments 5: Introduction: This is a bit brief, and it would be helpful to expand upon the relevant literature.
Response 5:Thank you very much for your comments, we have added this part to the third paragraph of the introduction.
Comments 6: Methods, Line 70: How many patients were excluded because of poor image quality and how many were excluded because of the lack of MR images?
Response 6: Thank you very much for your comments. Eight children could not hold their breath well resulting in poor image quality and 39 patients did not undergo MRI without images. Therefore, these 47 patients were excluded.
Comments 7: Methods, Line 74: “cystantin C”: Do you mean Cystatin C?
Response 7: Thank you very much for your comments and we apologized for any reading discomfort. Yes, we mean “Cystatin C”.
Comments 8: Methods, Line 96: The MEST-C score is introduced here. Please clarify what it represents, as it is related to the Oxford classification.
Response 8: Thank you very much for your comments. Mesangial hypercellularity (M), endothelial hypercellularity (E), segmental sclerosis or adhesion (S), interstitial fibrosis or tubular atrophy (T) and crescent (C) score. Concerned that readers would not understand exactly what Oxford typing means, we have already described the meaning of each MEST-C representation in the second paragraph of the introduction section.
Comments 9: Methods, Line 164: “differentiating M0 from M1, S0 from S1, and T0 from T1/2”. Please define these terms or refer to Table 2.
Response 9: Thank you very much for your comments. We apologized for any inconvenience this section may have caused you. ROC analyses with the AUC were used to evaluate the diagnostic performance of MRI parameters for differentiating pathological grading. However, because table 2 showed that only the M, S and T parameters differed between groups. So we only assessed the ability of the MRI parameters to discriminate between M, S and T grading.
Comments 10: The figure includes both MRI and histology images, not just MRI images.
Response 10: Thank you very much for your comments. We've made the changes as you suggested.
Comments 11: Figure 2: It would be helpful to show p-values for each of A-E.
Response 11: Thank you very much for your comments. We've made the changes as you suggested
Comments 12: Results, Line 220: “ADCMedulla generated the highest AUC of 0.791 (95% CI 0.633–0.949) for differentiating S0 from S1, and the AUC for differentiating M0 from M1 (AUC 0.833, 95% CI 0.611-1) and T0 from T1/T2 (AUC 0.774, 95% CI 0.521-1).” This wording is confusing. While ADCMedulla performed best for distinguishing S0 from S1, the wording makes it seem that ADCmedulla was also the best for the other two categories.
Response 12: Thank you very much for your comments. We apologized for any misunderstanding caused by our statements here. We've modified this section by breaking it down into two separate statements. ADCMedulla generated the highest AUC of 0.791 (95% CI 0.633–0.949) for differentiating S0 from S1. In addition, ADCMedulla had the AUC in differentiating M0 from M1 (AUC 0.833, 95% CI 0.611-1) and in differentiating T0 from T1/T2 (AUC 0.774, 95% CI 0.521-1).
Comments 13: Discussion, Line 307-315: “ADC values of kidney were higher in the high-risk group, which was consistent with the study of Dillman et al. [20].” “Children with lower ADC values indicated more renal microstructural changes, requiring a more targeted treatment regimen to minimize adverse outcomes.” In Figure 2, it seems like the high-risk group had lower ADC values.
Response 13: Thank you very much for your comments. We apologized for the mistake. It is true that the values were lower in the high-risk group, and the diffusive movement of water molecules was more restricted, suggesting more renal microstructural changes. Once again, we apologized for our careless mistake.
Comments 14: Discussion: It would be helpful to list future study directions.
Response 14: Thank you very much for your comments. We will continue to collect data on an ongoing basis and add further data in the future to validate our conclusions and conduct additional MRI studies of renal function in children. We will then further investigate the predictive value of various MRI sequences on the prognosis of patients with renal function to find the optimal sequence. Constructing practical and reliable clinical models to provide guidance for clinical work and help early identification of high-risk patients for early intervention and treatment.
Comments 15: Discussion: Would you say that one advantage of this study is that you can obtain prognostic information using imaging and a clinical factor without having to do a biopsy to get histological information, which in turn are variables that constitute the “international IgAN prediction tool at biopsy – adults” (https://qxmd.com/calculate/calculator_713/international-igan-prediction-tool-at-biopsy-pediatrics)? This could be emphasized more in the Discussion.
Response 15: Thank you very much for your comments. This point you make did not occur to us, and indeed it is as you say. In the future, our study is expected to further update the clinically recognized prediction formulas including the International IgA Nephropathy Prediction Tool, so that the histopathological parameters therein can be replaced by functional MRI parameters.
Comments 16: Discussion: It seems like the goal of the study was to differentiate between high-risk and low-risk groups. However, it might have also been useful to analyze direct clinical outcomes. How good of a gold standard is the International IgAN prediction tool at biopsy?
Response 16: Thank you very much for your comments. The Model fit for this tool improved to 30.3% and 22.2%, and the AIC improved to 1206 and 1228, c-statistics improved to 0.74 and 0.68. Calibration during follow-up showed good agreement between predicted and observed risks. Calibration of 5-year risks also showed good agreement between predicted and observed risks.
For other specific values, please refer to this document: “Updating the International IgA Nephropathy Prediction Tool for use in children”

Round 2
Reviewer 2 Report
Comments and Suggestions for Authors
Accept
Reviewer 3 Report
Comments and Suggestions for Authors
Thank you for making the improvements to the paper, which have substantially improved it. A few minor comments.
Comment 6: Thank you for the clarification. I would add this information to the manuscript.
Table 1: There is a typo “24 hhour-urinary protein”